# The Variance of Photophysical Properties of Tetraphenylethene and Its Derivatives during Their Transitions from Dissolved States to Solid States

**DOI:** 10.3390/polym14142880

**Published:** 2022-07-15

**Authors:** Ming Fang, Wenjuan Wei, Ruoxin Li, Liucheng Mao, Yuanheng Wang, Yan Guan, Qiang Chen, Zhigang Shuai, Yen Wei

**Affiliations:** 1Key Lab of Bioorganic Phosphorus Chemistry & Chemical Biology, Department of Chemistry, Tsinghua University, Beijing 100084, China; wwenjuan20@163.com (W.W.); liruoxin@mail.tsinghua.edu.cn (R.L.); mlc19@mails.tsinghua.edu.cn (L.M.); 2MOE Key Laboratory of Organic OptoElectronics and Molecular Engineering, Department of Chemistry, Tsinghua University, Beijing 100084, China; wyh18@mails.tsinghua.edu.cn (Y.W.); zgshuai@tsinghua.edu.cn (Z.S.); 3Analytical Instrumentation Center of Peking University, Center for Physicochemical Analysis and Measurement in ICCAS, Beijing 100871, China; yanguan@pku.edu.cn; 4Laboratory of Plasma Physics and Materials, Beijing Institute of Graphic Communication, Beijing 102600, China; chenqiang@bigc.edu.cn

**Keywords:** aggregation-induced emission, sol–gel synthesis, tetraphenylethene, emission mechanism

## Abstract

The study of aggregation-induced emission luminogens (AIEgens) shows promising perspectives explored in lighting, optical sensors, and biological therapies. Due to their unique feature of intense emissions in aggregated solid states, it smoothly circumvents the weaknesses in fluorescent dyes, which include aggregation-caused quenching of emission and poor photobleaching character. However, our present knowledge of the AIE phenomena still cannot comprehensively explain the mechanism behind the substantially enhanced emission in their aggregated solid states. Herein, to systematically study the mechanism, the typical AIEgens tetraphenylethene (TPE) was chosen, to elucidate its photophysical properties, the TPE in THF/H_2_O binary solvents, TPE in THF solvents depending on concentration, and the following direct conversion from a dissolved state to a precipitated solid state were analyzed. Moreover, the TPE derivatives were also investigated to supply more evidence to better decipher the generally optical behaviors of TPE and its derivatives. For instance, the TPE derivative was homogeneously dispersed into tetraethyl orthosilicate to monitor the variance of photophysical properties during sol–gel processing. Consequently, TPE and its derivatives are hypothesized to abide by the anti-Kasha rule in dissolved states. In addition, the factors primarily influencing the nonlinear emission shifting of TPE and its derivatives are also discussed.

## 1. Introduction

Since 2001, the terminology of aggregation-induced emission (AIE) was specifically cast by Tang et al. [1] the value and potential of substantially enhanced emissions of organic compounds in aggregated solid states started to gain importance, especially for their advantages in overcoming the critical weaknesses of dyes including the aggregation-caused quenching (ACQ) of emission and the poor photobleaching characteristics [2]. Up to now, AIE luminogens (AIEgens) have been widely utilized in the fields of lighting [3,4,5], optical sensors [6,7,8,9,10,11,12,13], biological therapies [14,15,16,17,18,19,20] and so on [21,22,23,24,25,26]. Hence, it shows the necessity to comprehensively understand the photophysical properties of AIEgens. To this purpose, great effort has been donated to reveal the emission mechanism of AIEgens.

For the investigations of the emission mechanism of AIEgens, the initial pioneer work by Tang et al. summarizes the optical behaviors of silole derivatives [1] and arylbenzenes [27]. The restriction of the intramolecular rotation (RIR) mechanism is applied to address the enhanced emission phenomena by adding water to the AIEgens solutions. In addition, the result based on the adjustments of viscosity and temperature of AIEgens solutions also supports the RIR mechanism. When the molecular structures of AIEgens lack the rotatable components such as 10,10′,11,11′-tetrahydro-5,5′-bidibenzo[a,b][7]annulenylidene and 5,5′-bidibenzo[a,b][7]annulenylidene, the RIR mechanism becomes difficult in explaining the restriction motions of molecules [28]. Then, the restriction of the intramolecular vibration (RIV) mechanism compensated for the RIR mechanism to establish the general principle of restriction of intramolecular motions (RIM). In 2006, Qian Peng and Zhigang Shuai et al. investigated the photophysical properties of two compounds, *cis,cis*-1,2,3,4-tetraphenyl-1,3-butadiene and 1,1,4,4-tetraphenyl-butadiene which demonstrated opposite optical behaviors [29]. One compound presents no emission in a dissolved state but substantially enhanced emission at low temperatures or the material showing intense emission at an aggregated solid state. Yet, another compound reversely demonstrates intense emissions in a dissolved state but no emissions in a solid state. The result brings them to the conclusion that the low-frequency phenyl ring twist motions and their Duschinsky mode mixing play crucial roles in supplying the nonradiative decay channels to quench emission. In 2013, Jong W. Chung and Soo Y. Park et al. mentioned the conformation switch of 2,3-bis(4′-methylbiphenyl-4-ly)acrylonitrile (CN-MBE) changing from (*E*)-CN-MBE to (*Z*)-CN-MBE by heating or UV irradiance [30]. They assigned the conformation-induced intense emission to the altered π–π intermolecular interaction. In addition, the investigations based on AIEgens involving excited-state intramolecular proton transfer (ESIPT) also display analogous impacts on photophysical properties. For instance, in 2013, Toshiki Mutai et al. [31] observed the faint dual emission of 2-(2′-Hydroxyphenyl)imidazo [1,2-α]pyridine and its derivatives in solutions. However, the materials gained intense emissions in solid states attributed to the emission mechanism switch from the normal to ESIPT fluorescence modes. In 2020, Michael Dommett et al. [32] analyzed the photophysical properties of 2′-hydroxchalcone derivatives and their mono-aryl analogs based on 2-hydroxyphenylpropenone. They also emphasized the critical role of efficient ESIPT in maximizing the quantum efficiencies of AIEgens in solid states. Very recently, Qian Peng and Zhigang Shuai cast out the new understanding of emission mechanisms of a majority of AIEgens that the intramolecular electron-vibration coupling plays a more crucial role than that of the intermolecular excitonic coupling [33]. Except for the explanations mentioned above, other mechanisms to explain the severe nonradiative decay of AIEgens include the switch from the bright (π,π*) state to the close-lying dark (n,π*) state [34], the quenching approaches through a conical intersection [35,36], the crystalline-induced quenching [37,38], the Herzberg–Tell vibronic coupling effect [39], and so on [2,40,41,42,43].

These mechanisms supply us with multi-parallel approaches to understanding the AIE phenomena, especially focusing on the illustration of the nonradiative quenching channels of AIEgens in dissolved states. However, the transitions of AIEgens from dissolved states to aggregated solid states do not attract enough attention which might hide more information that can reveal the mechanisms behind the enhanced emissions. Herein, by deviating from only addressing the nonradiative decay mechanisms, the attention is intentionally concentrated on monitoring the transitions of typical AIEgens involving tetraphenylethene (TPE) and its derivatives transforming from dissolved states to aggregated solid states. To explore the variance during the transitions, the photophysical properties of TPE in THF/H_2_O binary solvents, TPE in THF solvents with increasing concentration, and the followed direct transformation from a dissolved state to a precipitated solid state by directly evaporating solvent were cautiously characterized and discussed. The result displays the emission bands’ merging of TPE during the transition from a dissolved state to a solid state interpreted as the greater steric hindrance of TPE molecules and the hypothesized emission mechanism switch. Moreover, the tetraethyl orthosilicate (TEOS) was also employed to prolong the emission bands’ merging behavior of the TPE derivative during the transition from a dissolved state to a steric hindrance solid state for detecting more mid-states. Synergistically, the theoretic calculation results by Gaussian 16 W were also applied to discuss the energy transfer mechanism.

## 2. Materials and Methods

### 2.1. Materials and Synthesis

TPE (98%, Sigma-Aldrich, Shanghai, China), 1-methoxy-4-(1,2,2-triphenylethenyl)benzene (TPE-1, 98%, Tensus Biotech, Shanghai, China), 1-(4-bromophenyl)-1,2,2-triphenylethylene (TPE-2, 98%, Shanghai Aladdin Biochemical Technology Co., Ltd., Shanghai, China), 4-(1,2,2-triphenylvinyl)benzoic acid (TPE-3, 98%, Energy Chemical, Shanghai, China), absolute ethanol (EtOH, 99.8%, Sigma-Aldrich, Shanghai, China), tetrahydrofuran (THF, 99.9%, Sigma-Aldrich, Shanghai, China), tetraethyl orthosilicate (TEOS, 99%, Energy Chemical, Shanghai, China), and benzo[g,p]chrysene (98%, Toko Chemical Industry Co., Ltd., Shanghai, China) were utilized without further purification. The molecular structures of TPE, TPE-1, TPE-2, and TPE-3 are demonstrated in Figure 1a.

L-TPE denotes the TPE (fully dried powder) dissolved in THF solvent while L-TPE-*y* (*y* = 1,2, and 3) represents the corresponding TPE-*y* (fully dried powder) dissolved in ethanol solvent. P-TPE-*y* (not fully dried powder) signifies the precipitated solid states to directly evaporate L-TPE-*y* at ambient temperature without any further heat treatment utilized to accentuate the discrepancy compared to that of TPE-*y*. S-TPE-1 means the prepared sol by dispersing TPE-1 in TEOS. The S-TPE-1 (*x* mg/mL, *x* = 10^–3^, 10^–2^, 10^−1^, 5, and 10) sols were prepared based on the doping ratio *x* mg of TPE-1:1 mL of TEOS:1 mL of EtOH, and then stirring (200 rpm, IKA^®^ RCT basic) under ambient environment for 24 h to remove most of EtOH solvent for the following characterizations.

### 2.2. Characterizations

^1^H nuclear magnetic resonance (NMR) spectra were measured using a JEOL JNM-ECA400 spectrometer, and Delta 5.3.1 was applied to collect NMR data. Room-temperature emission and excitation spectra were characterized by SHIMADZU RF-6000 spectrofluorometer with 1 nm of increment, 5 nm/5 nm bandpasses of emission/excitation side slits, and 600 nm/min of scanning speed setting. Temperature-dependent emission spectra were obtained by a HORIBA Nanolog^®^ spectrofluorometer with 1 nm of increment, 2 nm/2 nm bandpasses of emission side entrance/exit slits, 2 nm/2 nm bandpasses of excitation side entrance/exit slits setting, accompanied with Oxford DN2 sample holder to control sample temperature in 80–300 K. Time-resolution emission intensity decay curves were recorded by a HORIBA Deltaflex lifetime fluorometer. The above-mentioned characterizations of solutions were all manipulated by utilizing quartz cuvettes with a 1 mm light path length. A PerkinElmer Lambda 750 was applied to measure the UV–Vis spectra with 1 nm of data interval and 267 nm/min of the scanning speed setting. To record UV–Vis absorption spectra of S-TPE-1, the S-TPE-1 (10^–3^ and 10^–2^ mg/mL) were measured in the quartz cuvettes through a 10 mm light path length. For the samples with heavy dopant amounts, the S-TPE-1 (10^−1^ mg/mL) was operated in quartz cuvettes through 1 mm light path length while the S-TPE-1 (5 and 10 mg/mL) were manipulated by coating on the quartz plates. The high-performance liquid chromatography (Agilent 1260 Infinity LC) was utilized to confirm the purity of commercial TPE and benzo[g,p]chrysene, and the results supported the high purities of the tested chemical reagents with single-peak signal, as presented in Appendix A.

The density functional theory (DFT) was employed to optimize the molecular structure in the ground state (S_0_) while the geometry of the excited states was calculated with the time-dependent density functional theory (TD-DFT). The calculations were operated at the tda/b3pw91 or m06-2x/6-31g(d,p)/scrf=(iefpcm,solvent=thf) level of theory by Gaussian 16 package [44]. To calculate the TPE in crystal, we performed quantum mechanics and molecular mechanics (QM/MM) with the tda/b3pw91 or m06-2x/6-31g(d,p) for QM and mechanics/uff for MM with the Gaussian 16 package.

## 3. Results and Discussion

The typically established theory [1,27] to check the AIEgens is to add the poorly dissolved solvent into the AIEgens’ solutions whose emissions are faint before the addition of the poorly dissolved solvent, consequently with the noticeably enhanced emission intensity as the evidence of confirming the AIE character, but there are considerable factors influencing the optical properties of obtained aggregated AIEgens. Such an intense emission is generally bridged to possible triggers such as aggregation [1,18,27,40], twisted intramolecular charge transfer [45,46,47,48], and molecular structure transformation [24,49,50,51]. However, the impact of solvent polarization on their photophysical properties does not arouse enough attention during the transitions from dissolved states to solid states. In fact, multiple investigations have mentioned the vital role of solvent polarization on the photophysical properties of AIEgens [48,52,53,54,55]. Therefore, the reasonable assessment method of the photophysical properties of AIEgens is necessarily designed specifically to distinguish the impacts of solvent polarization and the aggregation of molecules.

### 3.1. Photophysical Properties of TPE in Binary Solvents

Figure 2a displays the emission spectra of TPE in THF/H_2_O binary solvents recorded under 280 nm UV irradiance. The stated samples of TPE in THF/H_2_O binary solvents were prepared by mixing a 100-*f_w_*:*f_w_* (*f_w_* = water fraction (%)) volume ratio of TPE solution (TPE in THF, 0.1 mg/mL) and distilled water. With *f_w_* less than 70%, the emission spectra display the emission center (λ_cen_) at ~380 nm while the λ_cen_ moves to ~475 nm with a *f_w_* larger than 80% resulting from the aggregation-induced emission [1,27,56]. To gain more information during the transition, the photophysical properties of L-TPE depending on concentration and the following direct conversion from the dissolved state (L-TPE) to the precipitated solid state (P-TPE) were also examined.

### 3.2. Photophysical Properties of TPE in THF Solvent

Figure 3 shows the excitation-dependent three-dimension (3D) and two-dimension (2D) emission spectra of L-TPE with the concentration raising from 1 to 10 mg/mL, and the corresponding excitation spectra are displayed in Appendix A. The photos of TPE in THF solvent with a concentration of 10 mg/mL contained in a quartz cuvette (1 mm excitation light path length) under daylight and 280 nm irradiance (the same measurement setting for the emission spectra) are presented in Appendix A to exhibit the situation of excitation beam through the cuvette. The UV–Vis absorption spectra of L-TPE are demonstrated in Figure 2b with the first absorption band λ_cen_ = 315 nm and the absorption edge around 400 nm. For L-TPE (1 mg/mL), the excitation wavelength-dependent emission spectra (Figure 3a) point out the spectra constructed on the multi-emission bands [57,58]. The emission spectra obviously contain at least two emission bands, one relatively more intense emission band λ_cen_ = ~390 nm and another less intense emission band λ_cen_ = ~487 nm. As the doping concentration is increased to 2 mg/mL, the emission λ_cen_ = ~487 nm oppositely becomes more intense than that of emission λ_cen_ = ~390 nm. Furthermore, the third emission band is discerned in the L-TPE (5 and 10 mg/mL) under the irradiance wavelength longer than 400 nm performing the slightly red-shifting. Zooming into the UV–Vis spectra of L-TPE in 300–600 nm range illustrated in Figure 2b, the extra weak absorption band is recorded with the absorption band λ_cen_ = 475 nm in the samples of L-TPE (5 and 10 mg/mL). In addition, ^1^H NMR measurements were applied to examine the change of TPE in CDCl_3_ solvent (5 mg/mL) after the same UV irradiance treatment during the emission characterization of L-TPE (5 mg/mL). In Appendix A, the divergence cannot be detected, proving the good stability of TPE through the measurements of photophysical properties at high concentration with the short time of low strength UV irradiance; however, the analogous measurement of L-TPE with low concentration (0.1 mg/mL) and long duration of intense 280 nm UV irradiance was also studied which will be discussed in Section 3.5. Here, the low strength UV irradiance is to expose the sample under UV irradiance from SHIMADZU RF-6000 spectrofluorometer with 5 nm bandpass of excitation side slit while the intense UV irradiance means to expose the sample under the UV irradiance with 20 nm bandpass of excitation side slit setting. Then, the three noticed emission bands are approximately recognized including λ_cen_ = ~390 nm, ~487 nm, and ~530 nm with the soaring concentration of L-TPE in 1–10 mg/mL range. Combining the results of TPE in THF/H_2_O binary solvents with two emission centers including one performing λ_cen_ = ~380 nm (*f_w_* less than 70%) and another aggregation-induced emission center λ_cen_ = 475 nm (*f_w_* larger than 80%), the emission band λ_cen_ = ~390 nm is interpreted to the recombination transition from the higher excited state of (S_n_) of TPE, emission λ_cen_ = ~487 nm is assigned to the recombination transition of first excited state (S_1_) of TPE, and the interaction among TPE molecules probably involving electronic transitions at high concentration results in the slight red-shifting of emission band λ_cen_ = ~530 nm [2,33,59,60] which will be also discussed in Section 3.5 along with the theoretic calculated result.

### 3.3. Conversion of TPE Transforming from a Dissolved State to a Solid-State

The photos showing the performance of the direct conversion of L-TPE (10 mg/mL) transforming from a dissolved state to a precipitated solid state under 365 nm UV irradiance are demonstrated in Figure 1b-1–b-4. The faint cyan emission of L-TPE (10 mg/mL) is almost visually negligible compared to that of TPE with the overwhelming blue emission in a solid state. With the gradual evaporation of THF solvent, the visually distinguishable blue and cyan mixture emissions are observed during the transition from L-TPE to P-TPE. Subsequently, the resulting P-TPE was further treated in an oven at 70 °C and the corresponding emission spectra based on heat treatment time are collected in Figure 4a. Since the THF solvent evaporates quickly, the transition states from L-TPE to P-TPE are difficult to be distinguished from the recorded emission spectra (Figure 4a). The emission spectra all show one emission band shape (λ_cen_ = 442 nm).

As addressed by the previously reported investigations regarding the emission mechanism of TPE and its derivatives, photocyclization was applied to explain the multi-emission bands in solutions [50,61,62]. In solvents, the energy cost of TPE and its derivative during the transitions from excited states to ground states were assigned to the accommodation of the rotations and twisting of the benzene rings. Furthermore, the structures of the molecules transform into cyclic intermediates [50]. In solid states, the molecules cannot undergo such transformation under UV irradiance because of the very high energy requirement [50]. Contrastively, this manuscript comes up with the hypothesis that the TPE in a dissolved state abides by anti-Kasha’s rule. To supply some experimental evidence to estimate the photocyclization and the hypothesis, the emission and excitation spectra of benzo[g,p]chrysene (one of the potential intermediate states of L-TPE under UV irradiance) in THF solvents (1 and 5 mg/mL) and solid state are demonstrated in Appendix A. Appendix A presents the photos of benzo[g,p]chrysene in THF solvent with a concentration of 5 mg/mL under daylight and 300 nm UV irradiance (the same measurement setting for the emission spectra) to exhibit the excitation beam through the cuvette. The benzo[g,p]chrysene performs the analogous photophysical properties in THF solvents and solid states both with the similar curve shapes of excitation independent emission spectra. The emission center of benzo[g,p]chrysene in a solvent is located at around 411 nm (1 mg/mL) with the narrowed emission band at the higher concentration (5 mg/mL), and then the emission center moves to 418 nm in an aggregated solid state. Yet, other research demonstrates another conformation termed cyclized-TPE (molecular structure demonstrated in the next content, Section 3.5) with the emission center around 375 nm fairly close to the presentation of the emission center around 380–390 nm of L-TPE [50,61,62]. The comparison to the photocyclization will go on within Section 3.5 after the expanded characterizations on the TPE derivatives and the theoretic calculation result of TPE.

### 3.4. Photophysical Properties of TPE Derivatives

For the TPE derivatives, the excitation-dependent 3D and 2D emission spectra of L-TPE-1 (Appendix A), L-TPE-2 (Appendix A), and L-TPE-3 (Appendix A) all demonstrated the excitation dependent emission spectra. Furthermore, photos of the analogous direct conversions of L-TPE-1 (5 mg/mL in EtOH), L-TPE-2 (2 mg/mL in EtOH) and L-TPE-3 (5 mg/mL in EtOH) were also recorded in Figure 1(c-1–e-4). All the samples perform a similar emissive behavior with visually negligible emissions in dissolved states, but significantly intensified cyan emissions in solid states. In Appendix A, the emission and excitation spectra of TPE-2 and TPE-3 both show the excitation wavelength independent emissions centered at 454 and 463 nm, individually. In addition, the P-TPE-1 was also processed in an oven at 70 °C as another example of the heat treatment dependent emission in Figure 4b. In Appendix A, the parallelly detected XRD patterns of P-TPE-1 state the diffraction signals derived from the multi-crystal TPE-1 before and after 150 min of heat treatment. A continuous blue-shifting of the maximum emission center of P-TPE-1 is observed throughout the heat treatment processing assigned to the declined polarization of less EtOH residual [48,52,53]. Thus, the residual solvent may play a crucial role in the photophysical properties of the aggregated solid states of AIEgens. The L-TPE-1 (5 mg/mL in EtOH), L-TPE-2 (2 mg/mL), and L-TPE-3 (5 mg/mL) present the similar optical behavior with L-TPE, while the P-TPE-1 performs in accordance with P-TPE [4,33,63,64]. Thereby, the photophysical properties of TPE, TPE-1, TPE-2, and TPE-3 can be approximately considered the same in dissolved states or solid states.

In comparison between L-TPE-1 and P-TPE-1, despite the evidence revealing the recorded emission bands emerging after the transition, the mid-states still cannot be discerned. Since the TPE-1 possesses the advantage over TPE to be dissolved into EtOH solvent, it can be introduced into the typical sol–gel system based on TEOS precursors, TPE-1 was homogeneously dispersed into the TEOS to retard the altering process in order to visualize the whole transforming revolution. The UV–Vis spectra of S-TPE-1 with varied dopant amounts are presented in Figure 5. According to the normalized absorbance spectra (Figure 5), the results imply that with a rising concentration of absorption, the bands appear to be unchanged. The recognized discrepancy is a gradual red-shifting of the maximum absorption wavelength of the first absorption band [65,66]. For S-TPE-1 (10 mg/mL), the absorbance in the visible range may be triggered by the interaction among TPE molecules involving electronic transition or slightly attenuated transparency. Then, the absorption edge (zero-phonon) of the dispersed TPE-1 in sol is identified at around 400 nm (25,000 cm^−1^) [65,67].

Figure 6a,b shows the excitation-dependent 3D and 2D emission spectra of S-TPE-1 (10 mg/mL), respectively, inserted with the corresponding photos taken under daylight and 365 nm UV irradiance. The drop of as-prepared sol under daylight shows the qualified transparency and faint cyan emission under 365 nm UV irradiance. The S-TPE-1 performs in an analogous optical way with L-TPE-1. Adjusting the excitation wavelength in the 260–320 nm range, a sharp band centered at 407 nm and another faint emission band located at around 443 nm under the excitation wavelength in the range of 370–400 nm are discerned. To accelerate the sol–gel process, the sample was processed in the oven under 70 °C for 30 min to form the solid-state gel with the corresponding excitation-dependent 3D and 2D emission spectra given in Figure 6c,d, respectively [68]. The UV–Vis absorption spectra detected in parallel based on heat treatment are shown in Appendix A. Although the UV–Vis absorption spectra (Appendix A) do not report the identified discrepancy during the heat treatment, the sample gains the bright cyan emission under 365 nm UV irradiance (Figure 6c). Additionally, the emission centers gradually move from 407/443 to around 500 nm after the first 30 min of heat treatment. As the heat treatment time is prolonged to 60 and 120 min, the gel loses the flexibility to form the state like the glass, and the measured excitation wavelength-dependent 3D and 2D emission spectra are presented in Figure 6e,f and Figure 6g,h, respectively. The evolution tendency addresses the combination tendency of the multi-emission bands into one band which firstly red-shifts to around 500 nm (heat treatment for 60 min) and then slightly blue-shifts to around 480 nm (heat treatment for 120 min). The nonlinear tendency of emission center shifting principally stemds from the emission bands’ merging and the declined solvent polarization during the prolonged heat treatments [69,70].

To gain insight into the bands’ merging, the characterization of S-TPE-1 (10 mg/mL, 30 min of heat treatment in a solid state) was manipulated at 300–80 K under 300 nm UV irradiance which is demonstrated in Figure 7a. As the temperature drops from 300–80 K, the emission intensity centered around 480 nm gains a 6.2-fold improvement at 250 K which is significantly intensified to 67.6-fold and 130.4-fold at 200 K and 80 K, respectively. However, the emission intensity from the band centered at around 386 nm performs constantly in the 300–250 K range, and then reversely declines in the 250–80 K range. The corresponding time-resolution decay curves and parameters are reported in Figure 7b,c and Appendix A, respectively. The lifetimes monitored at 480 nm indicate the improvement from 1.886 ± 0.026 to 5.433 ± 0.026 ns in the 300–80 K range, and the faster lifetimes from the higher excited state monitored at 386 nm can be considered constant only varying from 0.545 ± 0.026 to 0.557 ± 0.026 ns in the 300–250 K range. Thereby, the successively prolonged lifetime by monitored at 480 nm addresses the restricted nonradiative decay approaches resulting from the dropping temperature contributing to the sharply boosted emission [1,27,29,61]. The weakened emission intensity of the band centered at 386 nm in the 250–80 K range highlights the restricted radiative transitions from higher excited states at lower temperatures which might enhance the internal conversion. Furthermore, it is noticed that the enhanced emission band slightly blue-shifts from 480 nm at 300 K to 450 nm at 80 K assigned to the temperature-dependent solvent polarization.

To further attest to the temperature-dependent emission center shifting, the emission spectra of the prepared S-TPE-1 (10 mg/mL), L-TPE-1 (5 mg/mL) and TPE-1 were all measured at 77 K. The emission spectra of the as-prepared sol at (I) RT, (II) 77 K and (III) one mid-state are presented in Figure 8a, attached with the corresponding photos. The measurement of mid-state (III) was operated after being frozen in liquid nitrogen (LN, 77 K) for 5 min and then exposed to the ambient environment for 10 s. The results display emission bands merging at 77 K. Furthermore, the emission band λ_cen_ = 513 nm (RT) shifts to 443 nm at 77 K. Consistent with the photos illustrated in Figure 8a, after taking out the S-TPE-1 from LN, the bright blue emission (λ_cen_ = 444 nm) intensity keeps declining with emission color turning to cyan (λ_cen_ = 472 nm). To extend the exposure time of the frozen S-TPE-1 (10 mg/mL) in the ambient environment for a total of 40 s, the emission intensity significantly decays to be consistent with the state before freezing treatment. The analogous phenomenon was also observed in L-TPE-1 (5 mg/mL) with two emission bands centered around 410 and 489 nm at RT but turning to a single-emission band centered around 452 nm at 77 K. The corresponding emission spectra are given in Figure 8b. Furthermore, the frozen L-TPE-1 (5 mg/mL, 77 K) also displays the rapid emission intensity decay as shown in the illustrated photos in Figure 8b after exposing the sample in an ambient environment for 20 s. Furthermore, the frozen L-TPE-1 returns to the state before treatment by exposing it to the ambient environment for a total of more than 40 s, whereas TPE-1 (fully dried powder) experiences neither the emission bands’ merging nor significant emission center shifting with the declined measurement temperature from 300 K to 77 K (Figure 8b). The emission only slightly performs blue-shifting from 449 to 445 nm. Accordingly, the restricted steric hindrance of the TPE-1 molecule is believed to account for the emission bands’ merging and the substantially intensified emission [1,71].

### 3.5. Theoretic Calculation and Discussion

According to the abovementioned results and analysis, the series of TPE and its derivatives perform analogous photophysical properties with negligible emission in dissolved states but significantly enhanced emission in aggregated solid states at RT. Additionally, the transitions from dissolved states to solid states cause the restricted nonradiative decay rate and the emission bands’ merging. Thus, to simplify the theoretic calculation and remove the minor impacts such as the functional groups from TPE-1 to TPE-3, the TPE was chosen as the candidate to display the general emission mechanism.

Figure 9 presents a portion of the calculated molecular orbitals of TPE in THF operated at the tda/b3pw91 or m06-2x/6-31g(d,p)/scrf=(iefpcm,solvent=thf) level of theory, and the parameters of the first three of calculated excited states are listed in Table 1. The calculated molecular orbitals based on the density functional setting including b3pw91 and m06-2x are quite similar. The HOMO and LUMO both display the intense contribution from the middle ethylene which forms the bridge to connect the contributions from adjacent four benzenes. However, the as-listed HOMOs (HOMO-6 to HOMO-1) and LUMOs (LUMO+1 to LUMO+5) illustrate the rare relationship with middle ethylene, the dominant contribution locating on the adjacent benzenes. It is consistent with other π-conjugation structures of AIEgens [72,73]. In Figure 10b and Figure 2b, the calculated UV–Vis absorption spectra obtained from the tda/b3pw91 or m06-2x/6-31g(d,p)/scrf=(iefpcm,solvent=thf) level of theories are both similar to the experimental results. Yet, the result calculated by the m06-2x density functional matches better, and the calculated UV–Vis absorption spectrum by b3pw91 density functional shows the slightly regular red-shifting. In experimental UV–Vis spectra, the first absorption band centered at around 315 nm and absorbed edge at around 400 nm (Figure 2b) is assigned to the S_0_ → S_1_ transition (calculated results with absorption band center/edge at 342 nm/420 nm for b3pw91 and 306 nm/380 nm for m06-2x, respectively) and the shoulder band centered at 285 nm is ascribed to the major donation from the S_0_ → S_3_ transition (Figure 10b and Table 1) which is overlapped with some contribution from absorption excitation transitions of the higher excited states [74]. The contributions from the calculated S_0_ → S_2_ transitions based on b3pw91 and m06-2x are very poor with the oscillator strength both less than 0.003.

Combining the results and analysis, the schematic diagram is supplied to display the possible dynamic process of L-TPE in Figure 10a. To explain the hypothesis of the anti-Kasha rule in L-TPE, the schematic energy transfer diagram of L-TPE is given in Figure 10c. For L-TPE (1 mg/mL), the emission mainly derives from the combination of the higher excited state of S_3_ → S_0_ (λ_cen_ = 390 nm, calculated λ_cen_ = 459 nm for b3pw91 and 415 nm for m06-2x). As the concentration increases to 2 mg/mL, the enhanced collisions among TPE molecules produce a larger population of the closing moment of TPE molecules which causes the slightly boosted π–π interaction and steric hindrance of TPE molecules, followed by the enhanced emission intensity from S_1_ → S_0_ (λ_cen_ = 487 nm in the experiment result, and λ_cen_ = 540 nm (b3pw91) or 507 nm (m06-2x) in the calculation, in Figure 3c and Figure 10b, respectively). With the concentration further increased to 5 and 10 mg/mL, the boosted collision and declined distance among TPE molecules result in more severe steric hindrance and generate the further strengthened emission from S_1_ → S_0_ (λ_cen_ = 487 nm, Figure 3e,g). Another slight red-shifting emission band (λ_cen_ = 530 nm) of L-TPE (5 and 10 mg/mL) under the excitation wavelength longer than 400 nm beyond the absorption S_0_ → S_1_ band should be the result of the interaction among TPE molecules involving electronic transitions under high concentration. The experimental and calculated UV–Vis state the absorption transition of the S_1_ → S_0_ edges at around 400 nm. Additionally, the extra faint absorption band centered around 475 nm is only recorded in the UV–Vis spectra of L-TPE (5 and 10 mg/mL, Figure 2b). We also notice that the absorbance value of the band (λ_cen_ = 475 nm) is very small reflecting the weak interaction among TPE molecules involving electronic transitions. Furthermore, since the calculated S_2_ state of TPE in THF solvent is close S_3_ state with an energy gap of 0.2 eV (M06-2x), it can lead to the fast internal conversion from S_3_ to S_2_ and S1 states. However, the strong nonradiative quenching of S1 excited states may be the reason that it results in the observed weak emission from the transitions of S_3_ → S_0_. Although the heavily doped L-TPE obtains enhanced emission to some degree, the samples still undergo the overwhelming nonradiative quenching decays to produce heat. After the transition from L-TPE to P-TPE, the four benzene groups of TPE molecules are all exposed under the strictly steric hindrance with surrounding TPE molecules, with the center distance between molecules around 6.509–10.147 Å in distance (Figure 10a) [75]. Accompanying the severe steric hindrance, the non-radiative recombination transition is tightly blocked with intensified emission intensity in orders of magnitudes. In Figure 10b, the calculated emission spectra of TPE in the crystal by the QM/MM calculation show the emission center at 472 nm for b3pw91 and 457 nm for m06-2x closing the experimental result at 449 nm, respectively. Then, to calculate the L-TPE and P-TPE, the m06-2x density functional offers a better opportunity to explicate the emission mechanism.

Previous investigations regarding the emission mechanism of TPE derivatives characterized by ultrafast time-resolved spectroscopy and computational studies [50,61,62,76] emphasize the vital role of triggering factors like rotation around the elongated middle-ethylene and the photocyclization accounting for the severe non-radiative recombination transition in solution. Since the cyclized-TPE with the emission center around 375 nm is fairly close to the presentation of the emission center around 380–390 nm of L-TPE [61], the time-dependent emission spectra of L-TPE in low concentration (0.1 mg/mL) under the intense 280 nm UV irradiance was cautiously characterized. In Figure 11, the deduced molecular interaction of L-TPE (0.1 mg/mL) under continuous long-term irradiance and the parallelly monitored ^1^H NMR (full spectra in Appendix A) and emission spectra are presented. The recorded ^1^H NMR spectra indeed reveal the gradual photocyclization of L-TPE to form the novel conformation (cyclized-TPE, as illustrated in Figure 11b) under continuous intense UV irradiance. In the ^1^H NMR spectra, the signals from TPE and cyclized-TPE primarily appear in the 7.0–7.1 ppm and 7.4–7.7 ranges, respectively [61]. In the corresponding emission spectra, the emission intensity centered at 380 nm is supposed to be the cyclized-TPE, but it fails to match the successively increasing tendency only with the emission shoulder centered at 320 nm following such tendency. So, the emission mechanism switch hypothesis can happen in parallel in L-TPE, and the emission signal from cyclized-TPE and S_3_ → S_0_ transitions of TPE are overlapped. In Figure 11c,d, the initially enhanced emission intensity during the irradiance in 1 min is interpreted as the excitation activated molecular vibration and rotation to weaken the intermolecular interaction among TPE molecules to some degree, consequently with the boosted transition from S_3_ → S_0_.

## 4. Conclusions

To analyze the transitions of TPE and its derivatives transforming from dissolved states to solid states, the emission spectra with multi-emission bands only perform in dissolved states but not in aggregated solid states. For TPE, the emission spectra obtained in THF/H_2_O binary solvents illustrate an emission band centered at ~380 nm disappearing with the percentage of water fraction larger than 80%. In addition, the high-performance liquid chromatography of TPE proves the high purity of the sample, displaying no impurity signal. Thereby, the emission band centered at ~380 nm of TPE in the solvent should not be assigned to the impurity. In this work, the hypothesis of emission mechanism switch is tried to address the emission bands’ merging phenomena to explain the disappeared emission band centered at ~380 nm. After the comparison with the theory of the UV-irradiance-induced photocyclization of TPE, it does not conflict with the hypothesis. Moreover, the calculated result by Gaussian 16 W also supports the hypothesis. So, the S_3_ → S_0_ and S_1_ → S_0_ transitions of TPE in dissolved states can be detected and the TPE in solid states only observes the S_1_ → S_0_ transition signal. The nonlinear emission shifting of TPE and its derivatives during the transitions from the dissolved states to solid states are interpreted as the disappeared solvent polarization (blue-shifting) and emission mechanism switch (red-shifting). In addition, similar photophysical properties have been found in TPE derivatives.

## Figures and Tables

**Figure 1 polymers-14-02880-f001:**
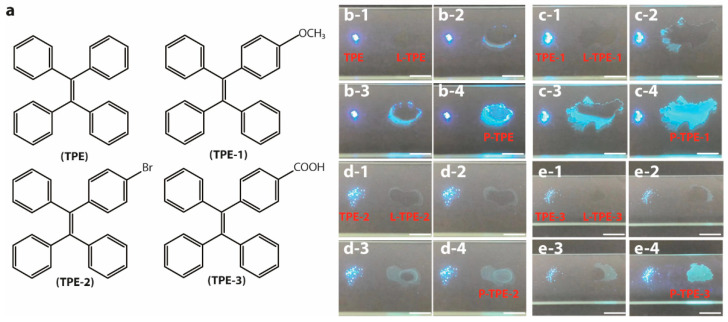
(**a**) Schematic molecular structures of TPE, TPE-1, TPE-2, and TPE-3. Direct conversion photos of (**b-1**–**b-4**) L-TPE (10 mg/mL), (**c-1**–**c-4**) L-TPE-1 (5 mg/mL), (**d-1**–**d-4**) L-TPE-2 (2 mg/mL), and (**e-1**–**e-4**) L-TPE-3 (5 mg/mL), transforming from the dissolved state to precipitated solid state (P-TPE-*y*) all compared to their corresponding TPE-*y* under 365 nm UV irradiance.

**Figure 2 polymers-14-02880-f002:**
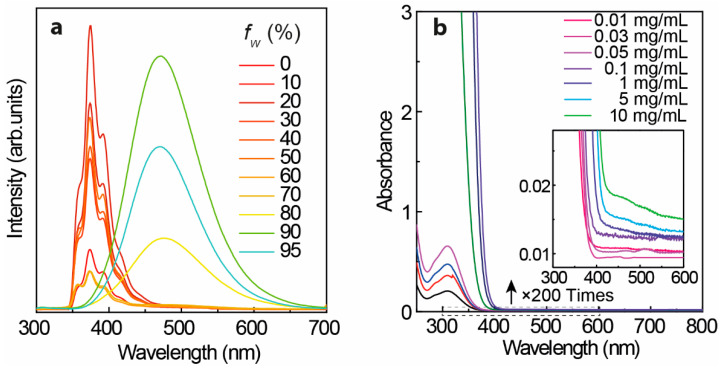
(**a**) Emission spectra of TPE in THF/H_2_O binary solvents under 280 nm excitation with varied addition of water volume fraction (*f_w_* (%)) into TPE solution (in THF, 0.1 mg/mL), and (**b**) UV–Vis spectra of TPE in THF solvent (all measurement operated in the quartz cuvettes through 1 mm excitation light path length).

**Figure 3 polymers-14-02880-f003:**
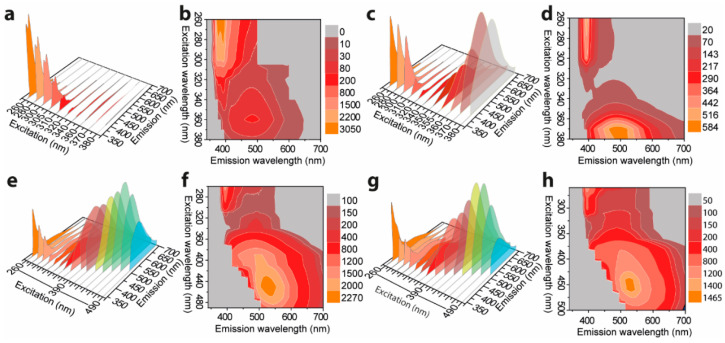
Excitation wavelength dependent three-dimension (3D) and two-dimension (2D) emission spectra of L-TPE: with (**a**,**b**) 1 mg/mL, (**c**,**d**) 2 mg/mL, (**e**,**f**) 5 mg/mL, and (**g**,**h**) 10 mg/mL of concentrations, respectively.

**Figure 4 polymers-14-02880-f004:**
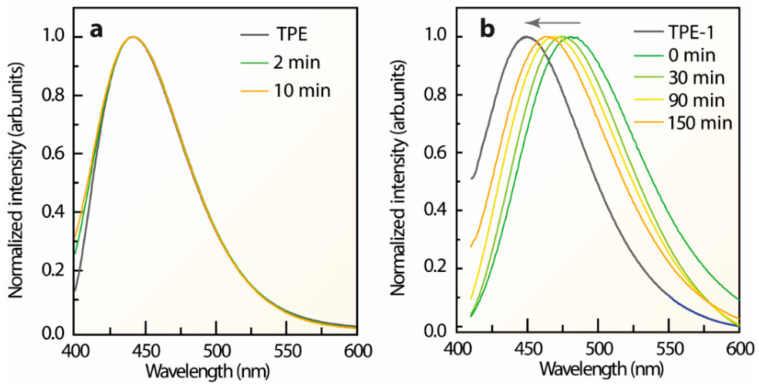
Emission spectra of (**a**) P-TPE and (**b**) P-TPE-1 solid states processed with 70 °C of heat treatments compared to that of the fully dried TPE and TPE-1 powders, respectively.

**Figure 5 polymers-14-02880-f005:**
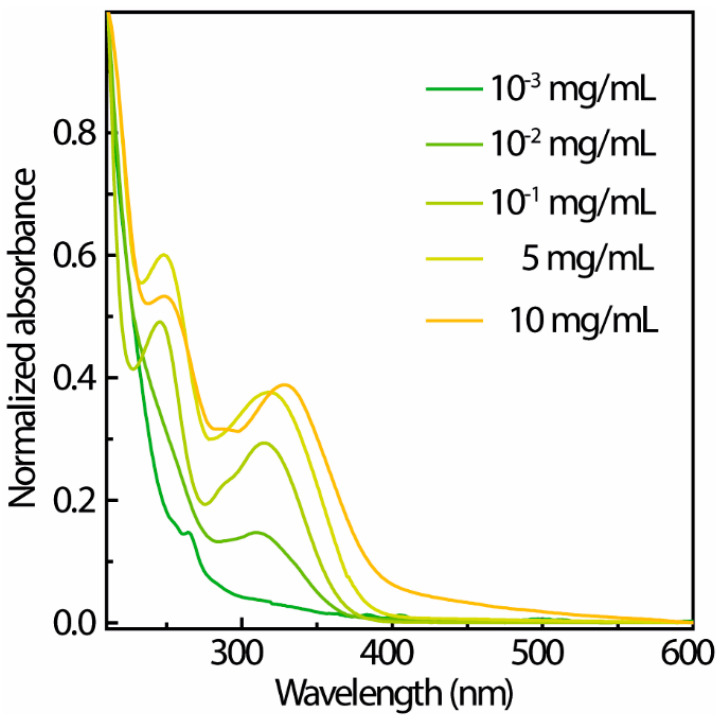
UV–Vis spectra of S-TPE-1 doped with varying concentrations.

**Figure 6 polymers-14-02880-f006:**
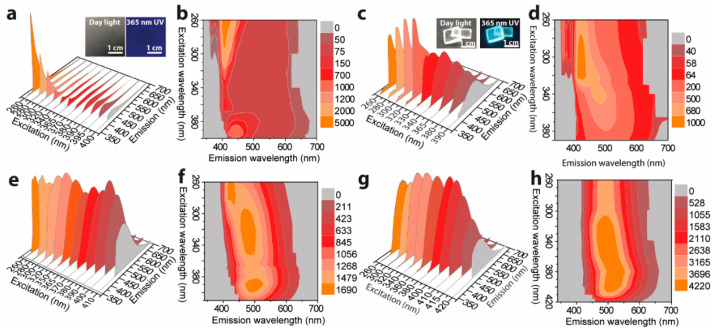
Excitation wavelength dependent 3D and 2D emission spectra of S-TPE-1 (10 mg/mL) processed with the 70 °C of heat treatment for (**a**,**b**) 0, (**c**,**d**) 30, (**e**,**f**) 60, and (**g**,**h**) 120 min.

**Figure 7 polymers-14-02880-f007:**
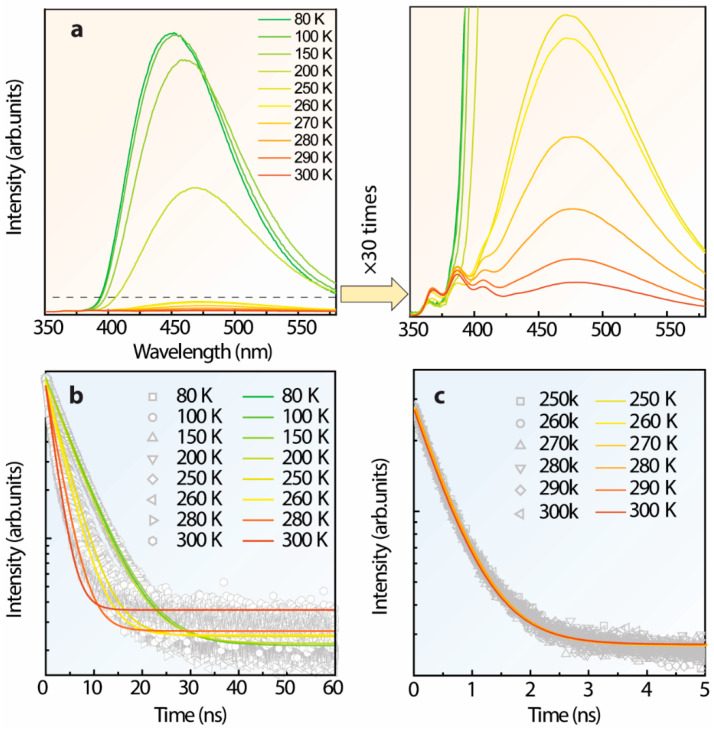
(**a**) Temperature-dependent emission spectra of S-TPE-1 (after 30 min of heat treatment under 70 °C, solid state gel), illustrated with the zooming area, (**b**,**c**) corresponding time-resolution intensity decay curves by monitoring the emission centered at 480 and 386 nm, respectively.

**Figure 8 polymers-14-02880-f008:**
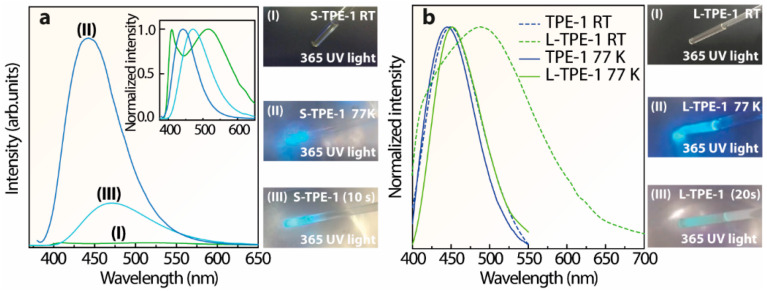
(**a**) Emission spectra of S-TPE-1 (10 mg/mL) measured at (I) RT, (II) 77 K (in LN), and (III) mid-state (after sinking in LN for 5 min and then exposing in the ambient environment for 10 s), accompanying with the corresponding photos, (**b**) normalized emission spectra of TPE-1 and L-TPE-1 (5 mg/mL) at RT and 77 K (in LN), illustrated with the photos of L-TPE-1 under 365 nm irradiance.

**Figure 9 polymers-14-02880-f009:**
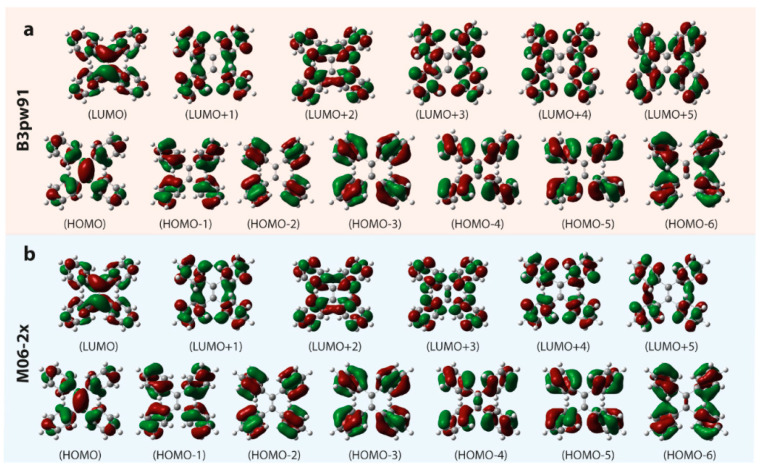
Electron density contours of the orbitals of TPE in THF solvent (contour level = 0.03) operated at the tda/(**a**) b3pw91 or (**b**) m06-2x/6-31g(d,p)/scrf=(iefpcm,solvent=thf) level of theory.

**Figure 10 polymers-14-02880-f010:**
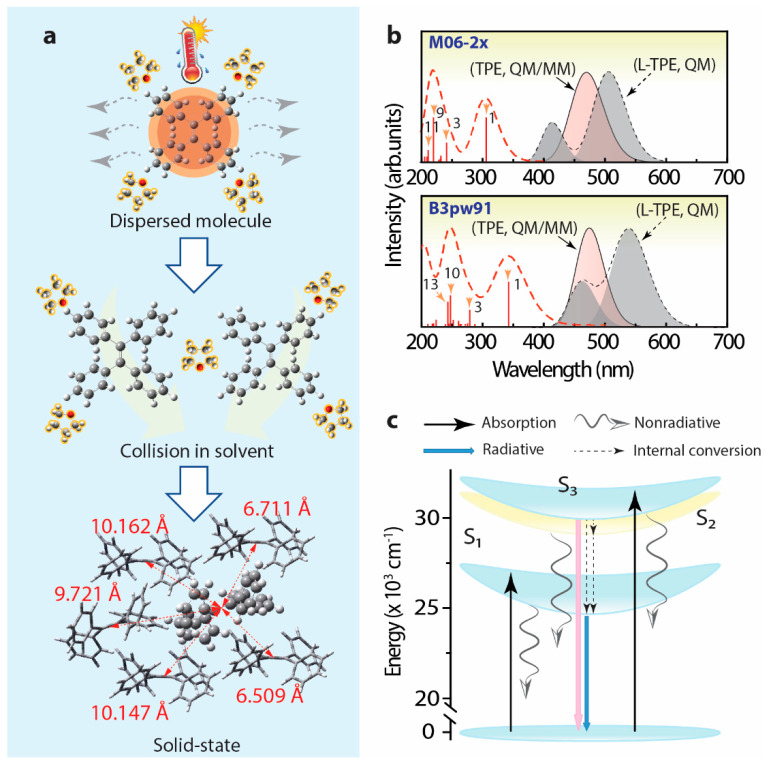
(**a**) Schematic diagram to display the interactions around TPE during the transition from a dissolved state to an aggregated solid state (the crystal structure data originated from CCDC 1521874 [74]), (**b**) calculated UV–Vis spectrum and emission spectrum of L-TPE based on TD-DFT calculations (QM, dash lines, emission spectrum summing up the separated S_1_ → S_0_, and S_3_ → S_0_ emission bands, halfwidth = 0.15 eV), as well as the calculated emission spectra of TPE in crystal based on QM/MM calculation (solid line, halfwidth = 0.15 eV), (**c**) schematic diagram to explain the energy transfer mechanism of TPE in a dissolved state.

**Figure 11 polymers-14-02880-f011:**
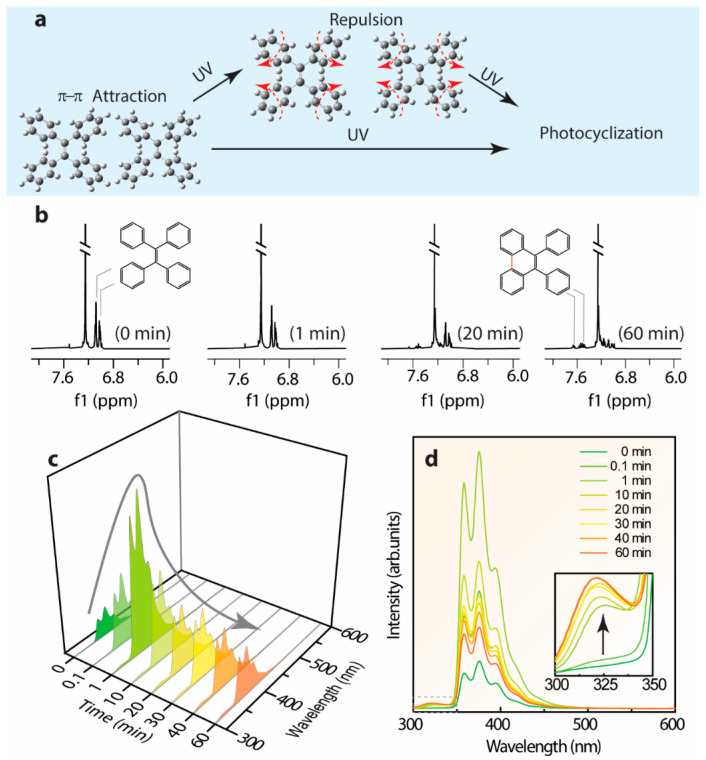
(**a**) The deduced molecular interaction of L-TPE (0.1 mg/mL) in low concentration under continuous long duration irradiance, and parallelly monitored time dependent (**b**) ^1^H NMR and (**c**,**d**) emission spectra.

**Table 1 polymers-14-02880-t001:** Parameters of the first three of calculated excited states.

Density Functional	Excited States	Transitions (Amplitude)	Energy Gap (eV)	Wavelength (nm)	Oscillator Strength
B3pw91	Excited state 1	HOMO → LUMO (0.972)	3.61	342	0.538
	Excited state 2	HOMO-5 → LUMO (0.034) HOMO-1 → LUMO (0.091) HOMO → LUMO+1 (0.832)	4.31	287	0.003
	Excited state 3	HOMO-6 → LUMO (0.064) HOMO-4 → LUMO (0.031) HOMO → LUMO+2 (0.886)	4.44	279	0.185
M06-2x	Excited state 1	HOMO → LUMO (0.970)	4.05	306	0.628
	Excited state 2	HOMO-5 → LUMO (0.028) HOMO-3 → LUMO+2 (0.035)HOMO-2 → LUMO+3 (0.030)HOMO-1 → LUMO (0.185) HOMO → LUMO+1 (0.637)	4.91	253	0.001
	Excited state 3	HOMO-6 → LUMO (0.077) HOMO-3 → LUMO+3 (0.025)HOMO → LUMO+2 (0.770) HOMO → LUMO+4 (0.029)	5.11	243	0.277

## Data Availability

The datasets generated during and/or analysed during the current study are available from the corresponding author on reasonable request.

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
