# Peer review of "The Variance of Photophysical Properties of Tetraphenylethene and Its Derivatives during Their Transitions from Dissolved States to Solid States"

_polymers, 2022, doi:10.3390/polym14142880_

Round 1

Reviewer 1 Report

This manuscript mainly reports the photophysical properties of tetraphenylethene (TPE) and its derivatives. Their emission behavior was evaluated in solutions with various concentrations, sol-gel matrices of tetraethylorthosilicate (TEOS), and cryogenic solutions. The authors mentioned that emission in dilute and concentrated solutions could be attributed to the S0 <- S3 and S0 <- S1 transitions. However, due to the confusing writing, the reviewer could not find the experimental details, and hard to follow the discussion. In addition, the topic of this research seems out of the scope of Polymers. Of course, the sol-gel reaction was applied to obtain an insight into the photophysical properties. Nonetheless, the topics of this manuscript entirely focused on the photophysics of TPEs. Therefore, the reviewer thinks that this manuscript should be resubmitted elsewhere after improving the discussion and writing.

1.         The composition of S-TPE-1, which is the mixture of TPE-1, ethanol and TEOS, was not found in the text, although the concentration appeared in the text, like 10 mg/mL. The reviewer highly recommends the authors should provide a graphical illustration of the preparation of the samples (L-TPE, P-TPE, and S-TPE).

2.         The effects of intermolecular interactions are confusing. The authors reported that the longer-wavelength emission in the concentrated solutions could be originated from some intermolecular interactions (Figure 3). Meanwhile, the detailed emission spectra in the sol matrices were provided only for 10 mg/mL. Therefore, readers could not evaluate the effects of intermolecular interactions. At least, emission behavior should be shown for S-TPE-1 with the other concentrations.

3.         The calculated S2 state is quite near the S3 state. Hence, the internal conversion from S3 to S2 should be very fast, resulting in the non-radiative quenching of the S3 state.

Author Response

Point-to-point responses

Reviewer 1:

This manuscript mainly reports the photophysical properties of tetraphenylethene (TPE) and its derivatives. Their emission behavior was evaluated in solutions with various concentrations, sol-gel matrices of tetraethylorthosilicate (TEOS), and cryogenic solutions. The authors mentioned that emission in dilute and concentrated solutions could be attributed to the S0→S3 and S0→S1 transitions. However, due to the confusing writing, the reviewer could not find the experimental details, and hard to follow the discussion. In addition, the topic of this research seems out of the scope of Polymers. Of course, the sol-gel reaction was applied to obtain an insight into the photophysical properties. Nonetheless, the topics of this manuscript entirely focused on the photophysics of TPEs. Therefore, the reviewer thinks that this manuscript should be resubmitted elsewhere after improving the discussion and writing.

Answer:

Dear reviewer, thanks for the comments and suggestions.

1. The composition of S-TPE-1, which is the mixture of TPE-1, ethanol and TEOS, was not found in the text, although the concentration appeared in the text, like 10 mg/mL. The reviewer highly recommends the authors should provide a graphical illustration of the preparation of the samples (L-TPE, P-TPE, and S-TPE).

Answer:

To better understand the TPE-y, L-TPE-y, and P-TPE-y, the initial Figure 1 and Figure S6 are combined to form the new Figure 1 in revised manuscript on Page 3. In addition, the definitions of TPE-y, L-TPE-y, and P-TPE-y are moved to Materials and Methods on Page 3. The preparation of S-TPE-1 sols can be found in Materials and Methods.

2. The effects of intermolecular interactions are confusing. The authors reported that the longer-wavelength emission in the concentrated solutions could be originated from some intermolecular interactions (Figure 3). Meanwhile, the detailed emission spectra in the sol matrices were provided only for 10 mg/mL. Therefore, readers could not evaluate the effects of intermolecular interactions. At least, emission behavior should be shown for S-TPE-1 with the other concentrations.

Answer:

For L-TPE (TPE in THF solvent with concentration of 5 and 10 mg/mL), the longer-wavelength emission (λcen = ~530 nm) can be observed under the excitation wavelength longer than 400 nm (Figure 3e to 3h). However, in the corresponding UV-Vis spectra (Figure 2b), the first absorption band edge of TPE is around 400 nm, along with much less absorption band centered at 475 nm. So, the longer-wavelength emission is interpreted to the weak intermolecular interactions which involve charge transitions. Please check the content on Page 5–6 (3.2. Photophyscial properties of TPE in THF solvent). For S-TPE-1 (10 mg/mL), the longer-wavelength emission is not observed.

3. The calculated S2 state is quite near the S3 state. Hence, the internal conversion from S3 to S2 should be very fast, resulting in the non-radiative quenching of the S3 state.

Answer:

For TPE in THF solvent, the calculated S2 state is near the S3 state which may lead the energy transfer to S2 and S1 states. However, the strong non-radiative quenching of S1 state is may the reason to result in the observed weak emission of the transitions from S3 to S0 states. To clear this point, the statement has been added on Page 13:“Furthermore, since the calculated S2 state of TPE in THF solvent is close S3 state with an energy gap of 0.2 eV (M06-2x) which may lead the fast internal conversion to S2 and S1 states. However, the strong nonradiative quenching of S1 state is may the reason to result in the observed weak emission from the transitions of S3→S0.”

Reviewer 2 Report

The authors have presented an investigation on the photophysical properties of tetraphenylethene and its derivatives during their transitions from dissolved states to solid states. The paper is well written and contains enough experimental results to qualify the paper to be accepted for publication in the Polymers Journal. However, I would like to suggest the following minor corrections before the paper can be published:

1-     The tittle needs to be modified e.g ‘’Investigations on the photophysical properties of tetra-2 phenylethene and its derivatives during their transitions from dissolved states to solid-states” or something similar

2-     The numerical values need to be added briefly in the abstract and the conclusion.

3-     There are a lot of grammatical errors in the manuscript e.g places like Line 41: “including such” and Line 162 “As such, we come up with a reasonable assessment method of the……….” e.t.c. As such, intense proofread is required

4-     Line 116: The stirring equipment and rpm used need to be included

Author Response

Point-to-point responses

Review 2:

The authors have presented an investigation on the photophysical properties of tetraphenylethene and its derivatives during their transitions from dissolved states to solid states. The paper is well written and contains enough experimental results to qualify the paper to be accepted for publication in the Polymers Journal. However, I would like to suggest the following minor corrections before the paper can be published:

Answer:

Dear reviewer, thanks for the comments.

1. The tittle needs to be modified e.g ‘’Investigations on the photophysical properties of tetra-2 phenylethene and its derivatives during their transitions from dissolved states to solid-states” or something similar

Answer:

The title has been modified to “The variance of photophysical properties of tetraphenylethene and its derivatives during their transitions from dissolved states to solid-states”

2. The numerical values need to be added briefly in the abstract and the conclusion.

Answer:

We compensate the conclusion with more detail information, please check on Page 15–16.

3. There are a lot of grammatical errors in the manuscript e.g places like Line 41: “including such” and Line 162 “As such, we come up with a reasonable assessment method of the……….” e.t.c. As such, intense proofread is required

Answer:

We have cautiously checked the grammatical errors.

4. Line 116: The stirring equipment and rpm used need to be included.

Answer:

The information of stirring equipment and rpm has been added on Page 3.

Reviewer 3 Report

The manuscript by Fang et.al. present the spectroscopic and theoretical (DFT calculations) study of TPE derivatives as a function of concentration and irradiation at 280 nm. The authors attribute the dual emission behaviour found for all the TPE-derivatives (Figure 3) to be from an anti-Kasha emission (from S3 excited state according to the TD-DFT data) and to aggregate emission. The hypotheses of the anti-Kasha behaviour should be carefully checked and supported by the experimental data since, contrary to what is expected, TPE in “diluted solutions” emits. If so it will be relevant to obtain the fluorescence quantum yields.

The manuscript is difficult to read and the anti-Kasha, the irradiance and the concentration dependence studies should be discussed in different sections. The anti-stokes behaviour is addressed using 0.1 mg/mL (3x10-4 M) solutions however it should also be explored in very diluted solutions (e.g. the 0.01mg/mL or 10-5 M). The fluorescence excitation spectra in Fig. S2 should be compared with the normalized absorption spectra since as is it does not offer support for the anti-Kasha behaviour. Moreover, the TPE absorption spectra of the 0.01 mg/mL and 0.1 mg/mL (fig 2b) solutions should be compared with the literature since the maxima at approximately 280 nm is not observed by others. Also going from the 0.01 mg/mL to the 0.1 mg/mL solution the ration between the 280 nm and 310 nm band changes! Is this only attributed to the formation of aggregates?

Does the steady-state and time-resolved fluorescence studies support the anti-Kasha emission centered at 390 nm? If the anti-kasha behaviour is present shouldn’t the emission intensity and decays times collected at the 390 nm band increase! The schematic diagram in Figure 10C does not illustrate the excited-state deactivation processes. In general, wavy arrows means, radiationless deactivations, so from this scheme we can assume that the main excited state deactivation channel in dissolved states is non-radiative transitions. Thus, where is the anti-Kasha emission and can the authors explain the radiative vs. radiationless deactivation from S3. In theory shouldn’t the radiative transition be favoured!

Can the authors also comment on the significant shift between the 280 nm absorption band and the 390nm emission band. Is this expected for an anti-Kasha emission!

What are the main conclusions of the 280 nm irradiation studies. Why the authors discarded the hypotheses of the 390 nm band to be from the photoproduct or other conformer. Did the authors computed the TD-DFT absorption spectra of the photoproduct?

Author Response

Point-to-point responses

Review 3:

The manuscript by Fang et al. present the spectroscopic and theoretical (DFT calculations) study of TPE derivatives as a function of concentration and irradiation at 280 nm. The authors attribute the dual emission behaviour found for all the TPE-derivatives (Figure 3) to be from an anti-Kasha emission (from S3 excited state according to the TD-DFT data) and to aggregate emission. The hypotheses of the anti-Kasha behaviour should be carefully checked and supported by the experimental data since, contrary to what is expected, TPE in “diluted solutions” emits. If so it will be relevant to obtain the fluorescence quantum yields.

Answer:

Dear reviewer, thanks for the comments. In fact, the emission signals assigned to the S3→S0 transitions of L-TPE all are very weak which is not suitable to measure the fluorescence quantum yields. Such as, the TPE derivative with four methyl substituents in THF solvent (TPE-4mM, J. Phys. Chem. A 2017, 121, 2572-2579) obtain a quantum yield of ~0.1%. In addition, if the concentration of L-TPE is too low, it will be very difficult to precisely measure the emission spectra by spectofluorometer. In addition, the TPE in THF solvent with concentration of 0.01mg/mL maybe result in the severe photobleaching.

1. The manuscript is difficult to read and the anti-Kasha, the irradiance and the concentration dependence studies should be discussed in different sections. The anti-stokes behaviour is addressed using 0.1 mg/mL (3x10-4 M) solutions however it should also be explored in very diluted solutions (e.g. the 0.01mg/mL or 10-5 M). The fluorescence excitation spectra in Fig. S2 should be compared with the normalized absorption spectra since as is it does not offer support for the anti-Kasha behaviour. Moreover, the TPE absorption spectra of the 0.01 mg/mL and 0.1 mg/mL (fig 2b) solutions should be compared with the literature since the maxima at approximately 280 nm is not observed by others. Also going from the 0.01 mg/mL to the 0.1 mg/mL solution the ration between the 280 nm and 310 nm band changes! Is this only attributed to the formation of aggregates?

Answer:

To improve the manuscript structure, Results and Discussion section has been divided into five parts including 3.1. Photophysical properties of TPE in binary solvents, 3.2. Photophysical properties of TPE in THF solvent, 3.3. Conversion of TPE transforming from a dissolved state to a solid-state, 3.4. Photophysical properties of TPE derivatives, and 3.5. Theoretic calculation and discussion.

The emission signals assigned to the S3→S0 transitions of L-TPE is very weak. The 0.1 mg/mL of TPE in THF solvent is utilized to investigate the photophyscial properties of TPE in binary solvent (H2O/THF). In addition, the TPE in THF solvent with concentration of 0.01mg/mL maybe result in the severe photobleaching.

The excitation spectra are to record the contribution of the scanned exciation wavelength irradiance for the monitored emission while the UV–Vis spectra are to record the absorbance of materials under the scanned wavelength irradiance. If the materials abide by the Kasha’s rule, the materials should have similar excitation and UV–Vis spectra.

The UV–Vis spectra of TPE in THF solvents with concentration in 0.01–1.0 mg/mL have been repeated. The repeated UV–Vis spectra present a weak shoulder absorption band centered at ~285 nm which matches with the result from reference Chem. Commun. 2014, 50, 12058. The content has been revised on Page 13. The TPE can be well dissolved into THF solvent which should not cause the aggregation of TPE molecules.

2. Does the steady-state and time-resolved fluorescence studies support the anti-Kasha emission centered at 390 nm? If the anti-kasha behaviour is present shouldn’t the emission intensity and decays times collected at the 390 nm band increase! The schematic diagram in Figure 10c does not illustrate the excited-state deactivation processes. In general, wavy arrows means, radiationless deactivations, so from this scheme we can assume that the main excited state deactivation channel in dissolved states is non-radiative transitions. Thus, where is the anti-Kasha emission and can the authors explain the radiative vs. radiationless deactivation from S3. In theory shouldn’t the radiative transition be favoured!

Answer:

In figure 7b and 7c, the time-resolution intensity decay curves of S-TPE-1 (10 mg/mL) by monitored at 480 and 386 nm confirm two different lifetime values which is consistent with the hypothesis of two emission bands derived from the transitions of varied excited states. In Figure 7a, the temperature dependent emission spectra display the restricted nonradiative decay approaches from S1→S0 transitions in 300–80 K range which results in the enhanced emission intensity and increased lifetime (1.886±0.026 to 5.433±0.026 ns). But the emission intensity of the band centered at ~386 nm performs constantly in the 300–250 K range, and then declines in the 250–80 K range. The corresponding time-resolution intensity decay curves match with the results in 300–250 K range with lifetime in 0.545±0.026 to 0.557±0.026 ns range. However, it is difficult to measure the time-resolution intensity decay curves by monitored at 386 nm in 250–80 K range. It is due to that the related emission intensity originated from transitions of higher excited states (λcen = ~386 nm) is much less than that of S1→S0 transition. The weakened emission intensity of band centered at 386 nm points out the restricted radiative transitions from higher excited states at lower temperature which might enhance the internal conversion. The more discussion has been added on Page 10.

The figure 10c has been improve with more information, please check on Page 14.

3. Can the authors also comment on the significant shift between the 280 nm absorption band and the 390nm emission band. Is this expected for an anti-Kasha emission!

Answer:

I am very sorry that I cannot make comments on this question. Based on my knowledge, I do not know the possible relationships between these factors.

4. What are the main conclusions of the 280 nm irradiation studies. Why the authors discarded the hypotheses of the 390 nm band to be from the photoproduct or other conformer. Did the authors computed the TD-DFT absorption spectra of the photoproduct?

Answer:

The time-dependent 1H NMR spectra of TPE in THF solvent with concentration of 0.1 mg/mL under 280 nm UV irradiance point out the continuous photocyclization of TPE to form the novel conformation of cyclized-TPE. However, the corresponding emission spectra do not match the successively increasing emission intensity. So, the emission mechanism switch hypothesis can parallelly happen in L-TPE. In this manuscript, we emphasize the possibility of S3→S0 transitions of TPE in dissolved states. Then, we compare the hypothesis with previous publications (References: Angew. Chem. Int. Ed. 2020, 59, 14903–14909, Chem. Sci. 2018, 9, 4662–4670, ACS Nano 2020, 14, 2090–2098, J. Phys. Chem. A 2017, 121, 2572–2579) which discussed the photoproduct or other conformer. The related discussion is on Page 14–15. Therefore, the main conclusion is that the emission signal from cyclized-TPE and S3→S0 transitions of TPE are overlapped, please check the revised content on Page 14–15. The theoretic studies of photocyclization of TPE and its derivatives can be found in the references Chem. Sci. 2018, 9, 4662, and J. Phys. Chem. A 2017, 121, 2572–2579 which are cited in this manuscript.

Round 2

Reviewer 1 Report

All queries have been addressed. I feel this manuscript is publishable in its current form.

Reviewer 3 Report

From my point of view the manuscript was not sufficiently improved and the anti-kasha behavior is not proved, to warrant publication in Polymers. In general, fluorescence from higher Sn (n > 2) can be observed when energy separation between the levels is favorable (necessary condition) and the Sn-S1 internal conversion is prohibited based on symmetry grounds (sufficient condition). However this was not proved in the revised manuscript.

Moreover:

“The manuscript by Fang et al. present the spectroscopic and theoretical (DFT calculations) study  of TPE derivatives as a function of concentration and irradiation at 280 nm. The authors attribute the dual emission behaviour found for all the TPE-derivatives (Figure 3) to be from an anti-Kasha emission (from S3 excited state according to the TD-DFT data) and to aggregate emission. The hypotheses of the anti-Kasha behaviour should be carefully checked and supported by the experimental data since, contrary to what is expected, TPE in “diluted solutions” emits. If so it will be relevant to obtain the fluorescence quantum yields.
Answer:
Dear reviewer, thanks for the comments. In fact, the emission signals assigned to the S3→S0 transitions of L-TPE all are very weak which is not suitable to measure the fluorescence quantum yields. Such as, the TPE derivative with four methyl substituents in THF solvent (TPE-4mM, J. Phys. Chem. A 2017, 121, 2572-2579) obtain a quantum yield of ~0.1%. In addition, if the concentration of L-TPE is too low, it will be very difficult to precisely measure the emission spectra by spectofluorometer. In addition, the TPE in THF solvent with concentration of 0.01mg/mL maybe result in the severe photobleaching.”

 R: Nowadays the determination of fluorescence quantum yields with values lower than 0.1% with reliability is possible!

 Answer:

In figure 7b and 7c, the time-resolution intensity decay curves of S-TPE-1 (10 mg/mL) by monitored at 480 and 386 nm confirm two different lifetime values which is consistent with the hypothesis of two emission bands derived from the transitions of varied excited states. In Figure 7a, the temperature dependent emission spectra display the restricted nonradiative decay approaches from S1→S0 transitions in 300–80 K range which results in the enhanced emission intensity and increased lifetime (1.886±0.026 to 5.433±0.026 ns). But the emission intensity of the band centered at ~386 nm performs constantly in the 300–250 K range, and then declines in the 250–80 K range. The corresponding time-resolution intensity decay curves match with the results in 300–250 K range with lifetime in 0.545±0.026 to 0.557±0.026 ns range. However, it is difficult to measure the time-resolution intensity decay curves by monitored at 386 nm in 250–80 K range. It is due to that the related emission intensity originated from transitions of higher excited states (λcen = ~386 nm) is much less than that of S1→S0 transition. The weakened emission intensity of band centered at 386 nm points out the restricted radiative transitions from higher excited states at lower temperature which might enhance the internal conversion. The more discussion has been added on Page 10.”

R: The Time -resolved fluorescence data (and not, "time-resolution decay curves") is not clear. How were the fluorescence lifetimes deconvoluted and fitted!

This manuscript is a resubmission of an earlier submission. The following is a list of the peer review reports and author responses from that submission.

Round 1

Reviewer 1 Report

This manuscript demonstrates that tetraphenylethene (TPE) and its derivatives change their emission properties during the transformation from a solution state to a solid-state. The experimental and theoretical studies imply that this class of materials shows emissions from both S1 and S3 states. These results are interesting and important for understanding the fundamental photophysics of TPE derivatives and developing functional luminescent materials. However, this research mainly focuses on photophysical properties of small molecules rather than polymer chemistry and appears to be suitable for other journals, such as Materials.

Reviewer 2 Report

The authors have studied the relation of aggregation-induced emission (AIE) effect in the solid states and upon dissolution.  While the topic would be of interest for Molecules readers, however, the text is very confusing and lost in many details. Summary points as taken home lessons are lacking. For instance, it is unclear a close pin-point between AIE  and ACQ effects function of the solvent concentration (see  Crystals 2021, . It is not clear how the non-solvent concentration affects Crystals 2020, 10(4), 269; https://doi.org/10.3390/cryst10040269

Reviewer 3 Report

Low molar mass TPE derivatives were examined to supply some evidence to better understanding of their optical behaviour. The paper is interesting, but it is not suitable for the journal in field of polymers. The described investigation do not provide some new information in field of polymers. The authors should submit the paper for journals in field of spectroscopy or optics.